# *Mentha piperita* L. Micropropagation and the Potential Influence of Plant Growth Regulators on Volatile Organic Compound Composition

**DOI:** 10.3390/molecules25112652

**Published:** 2020-06-07

**Authors:** Jacek Łyczko, Krystian Piotrowski, Kornelia Kolasa, Renata Galek, Antoni Szumny

**Affiliations:** 1Department of Chemistry, Wrocław University of Environmental and Life Sciences, ul. Norwida 25, 50-375 Wrocław, Poland; antoni.szumny@upwr.edu.pl; 2Department of Plant Breeding and Seed Production, University of Environmental and Life Sciences, pl. Grunwaldzki 24a, 50-363 Wrocław, Poland; krystian.piotrowski1995@gmail.com (K.P.); kornelia.kolasa@gmail.com (K.K.); renata.galek@upwr.edu.pl (R.G.)

**Keywords:** GC-MS, HS-SPME, micropropagation, OACs, peppermint, VOCs

## Abstract

Due to the industrial use of *Mentha piperita* L. (peppermint), it is important to develop an optimal method to obtain standardized plant material with specific quality parameters. In vitro cultures may allow the production of desirable odor-active compounds (OACs) and improve their share in the plant aroma profile. There are two types of explants that are commonly used, apical meristems and nodal segments. In this study, the best overall effects were shown to be produced by the combination of MS medium with the addition of 0.5 mg·dm^−3^ indolyl-3-butyric acid. In this case, a very high degree of rooting was found (97% for apical meristems, 100% for nodal meristems), lateral shoots were induced in 83% of both types of explant, and the content of OACs in the plant aroma profile increased significantly, especially menthofurolactone and *cis*-carvone oxide, responsible in this case for a characteristic mint-like aroma. The comparison of the volatile organic compounds (VOCs) obtained from plants of different origin by GC-MS showed no significant differences in their qualitative composition. Moreover, in-vitro-cultivated peppermint on a medium containing 0.5 mg·dm^−3^ 2-isopentinloadenine and 0.1 mg·dm^−3^ indolyl-3-acetic acid showed significant amounts of menthofurolactone in its VOC composition.

## 1. Introduction

Biotechnology tools such as in vitro cultures are useful for obtaining unified plant material with a guaranteed aroma quality, since the physical and chemical cultivation conditions, including temperature, photoperiod basal medium, the addition of plant growth regulators, and other factors such as CO_2_, may be strictly controlled [1,2,3]. In vitro cultures also allow for the cultivation of plants that do not contain undesirable chemical compounds [4]. This is especially important currently, when the use of chemical compounds naturally occurring in plants is becoming increasingly appreciated. The usable raw material of peppermint consists of herbs and leaves (fresh or dried), oil, or herb extract. In order to obtain the raw material, the plant is cut before flowering, because it contains the most active substances when harvested in this way.

Peppermint (*Mentha piperita* L.), is widely used in the cosmetics, pharmaceutical, and food industries [5,6,7,8]. One of its most valuable products is its essential oil (EO) [9]. The plant may contain about 300 different volatile constituents, mainly esters, ketones, and terpene oxides [10], but the most important is menthol [11], which belongs to the cyclic monoterpenic alcohols. It has three chiral centers in its structure, which allows it to exist in the form of eight stereoisomers. The naturally occurring isomer of menthol in peppermint is (−)-menthol, with a 1*R*, 2*S*, 5*R* configuration [12,13]. The levorotatory (−)-menthol isomer is responsible for the characteristic taste, aroma, and properties of menthol [14]. Furthermore, characteristic peppermint EO volatile organic compounds (VOCs), such as 1,8-cineole, limonene, *trans*-menthyl acetate, and *trans*-sabinene hydrate, are appreciated in the cosmetics and pharmaceutical industries for their refreshing aroma and bioactive features [15,16]. In addition, peppermint has been proven to have pharmacologically beneficial properties, such as being antioxidant, anti-allergic, antiviral, and antibacterial, preventing the development of microorganisms, and even inhibiting the development of cancer cells [5,13,17,18,19]. When used on small wounds, peppermint EO supports healing, making it an alternative to much more expensive lavender or tea tree EOs [15,20].

The food industry uses dried peppermint leaves in the form of mint teas or as blends for infusions. In this case, not only the EO composition is significant, but also the plant aromatic profile created by released volatile organic compounds (VOCs). As some previous studies have shown [21,22], both factors-EO and VOC composition-can have a significant impact on the plant’s material value regarding its product purpose. The EO is used for flavor and aroma purposes for chewing gums, hard candy, chocolate, liqueurs, or spirits. The tobacco industry also uses it to make menthol cigarettes [1]. Studies indicate that the aqueous extract of peppermint supports the defense mechanisms of radish (*Raphanus sativus* L.) and tomato (*Solanum lycopersicum* L.) against oxidative stress, and at the same time inhibits seed germination and weed seedling growth [13,23]. A similar inhibiting effect was observed on the germination of Mediterranean weeds after application of cinnamon, lavender, and peppermint EOs [24]. These observations might be the basis for developing a natural herbicide.

In vitro plant cultures offer the possibility of obtaining homogeneous biological material on a large scale. Knowledge of this topic might also enable better adaptation of technological lines, including in bioreactors. This information may be of special value today, when the world is once again seeking natural chemicals in which peppermint is abundant. The properties of plant growth regulators (PGRs) on the micropropagation of several herbs, including peppermint, have been evaluated in the context of their impact on growth and development; they can be used to obtain microseedlings from protoplasts, nodal and apical segments, and leaf discs for the direct or indirect (with proliferation of callus) method of morphogenesis, and the possibility of their use to increase compounds which are attractive for the bioeconomy has also been explored [1,19,25,26]. To optimize the regeneration process from different initial peppermint explants, PGRs such as BA (6-benzyloadeninopurine), BAP(6-benzyloaminopurine), TDZ (thidiazuron), KIN (kinetin), ZEA (zeatin) and 2.4D (2,4-dichlorophenoxyacetic acid), NAA (α-naphthylacetic acid), IAA (indole-3-acetic acid), or IBA (indolyl-3-butyric acid) have mainly been used, which all have different effects on morpho- and organogenesis, playing a major role in many functions [27,28,29,30,31,32,33,34,35,36,37]. Appropriate protocols for protoplast culture were developed by Sato et al. (1993) [27] and then Jullien et al. (1998) [38]. In both studies, the B5 medium was used as the basic element, but that used in the first one contained 1 mg/L NAA, 0.4 mg dm^−3^ BA, 0.5% sucrose, 0.5 M mannitol, and 0.1% Gelrite. Julien et al. (1998) [38] obtained the best results in the first stage of the division of protoplasts into free genotypes (“Mitcham Digne 38”, “Mitcham Ribecourt 19”, and ”Todd’s# × 2019”) by combining B5 with 1.0 µM 2.4-D, 2.5 µM NAA, and 4.0 µM BA. In the next stage, the addition of 2.3 μM thidiazuron allowed for increased bud formation and regeneration of shoots from calli with a frequency of over 50%. The authors noticed differences in regeneration capability depending on the genotype and the path of shoot regeneration. Additionally, liquid medium was more efficient in supporting first protoplast divisions, and the solid medium was clearly more suitable to support subsequent cell divisions leading to the formation of microcalli. Breeding protoplasts was also useful in the case of new hybrids [28,30]. Faure et al. (1998) [31] described the process of the in vitro organogenesis of peppermint shoots from explants derived from leaf disks. It was observed that the regeneration occurred within 6 weeks of establishing the culture. The best results were obtained when explants were grown on MS medium with the addition of 300 mM mannitol, 2.0 µM BA, and 2.0 µM IBA, and, after two weeks, the explants were transferred to a new medium with 0.5 µM NAA, 9.0 µM BA, and 0.5 µM TDZ (thidiazuron), which did not contain mannitol. In the case of peppermint, 78% regeneration was obtained, which may indicate the usefulness of this method for genetic transformation with the help of *Agrobacterium tumefaciens*.

An experiment for green mint (*Mentha spicata* L.) was performed by Samantaray et al. (2012) [32], where the effect of growth regulators on plant development was examined. The concentration of 2.5 mg/L BAP stimulated greater a yield of new shoots (10) and leaves (38) from the embryogenic callus at the morphogenesis stage, and the full regeneration of shoots was shorter, lasting 62 days. Experience has shown that similar results for root growth were achieved with the participation of IAA and NAA (4 mg·dm^−3^). The combination of NAA 0.1 mg·dm^−3^ and BA 2.0 mg·dm^−3^ was also repeated by Krasnyanski et al. (1998) [30] in studies on the somatic hybridization of mint for step protoplast division, and for the shooting of microcalli in combinations of BA (1–2 mg·dm^−3^) and TDZ (2–3 mg·dm^−3^). Moreover, a similar set of regulators with NAA (1 mg·dm^−3^) and BAP (2 mg·dm^−3^) as the best combination with a positive effect on the regeneration of shoots of peppermint from nodal segments was indicated by Sharan et al. (2014) [33]. An experiment conducted by Mehta et al. (2012) [34] showed that MS medium with 2.0 mg·dm^−3^ BAP was the best for starting growth and multiplying shoots. A positive response to shoot proliferation was also observed after adding 0.5–4.0 mg·dm^−3^ KIN (kinetin) to the BAP medium (0.5 mg·dm^−3^). The best results were obtained at 0.5 mg·dm^−3^ BAP and 3.0 mg·dm^−3^ KIN. Different rations of IBA were used for rooting, and the highest percentage of rooted plants was recorded on MS medium with 2.0 mg· IBA.

Over the years, the selection of growth regulators has been discussed in many experiments for peppermint, but the obtained results indicated a specific reaction of the chosen genotypes for particular experiments on plant tissue culture conditions. The aim of this study was to develop a rapid peppermint micropropagation protocol for a chemotype that originates from Torseed S.A., and to determine the effect of growth regulators on the chemical composition. We propose a new combination of cytokinin 2iP (2-isopentinloadenine) with auxin IAA (indole-3-acetic acid) for our purpose.

## 2. Results and Discussion

### 2.1. In Vitro Cultures

The conducted analysis of variance showed that the type of starting material was significant for most of the analyzed characteristics, except for the length of the largest lateral new shoots and the number of new lateral shoots. The mean values were grouped using Turkey’s test (Table 1). A significant influence of the medium was noted for features such as plant height, the length of the smallest lateral branches, and root length (Figure 1). Interactions between the medium and the type of explants were significant for four parameters, namely plant height, length of smallest new side shoots, number of new modal segments, and average root length (Table 1). No source of variance related to the medium, type of explant, or the interaction between them was statistically significant for the average number of new shoots and the length of the largest lateral new shoots.

Tukey’s test showed that the highest average plant height was obtained for plants cultivated on control media MS-K and MS-3 when the source of explants was apical meristems, while the remaining values obtained for explants were almost 50% lower. Generally, after four weeks of cultivation, taller plants for both types of explant used for study were observed on MS-K medium without PGR and with the addition of auxin–IBA at a concentration 0.5 mg·dm^−3^ (Figure 1), which was justified in the absence of cytokinins in the medium. Cytokinins generally inhibit the elongation of cells but promote their division in plant roots and shoots. They are involved primarily in cell growth and differentiation, but also affect apical dominance and axillary bud growth. In turn, auxins usually affect elongation growth and rooting and inhibit the development of lateral shoots. The balance between auxins and cytokinins plays a key role in favorable plant growth in tissue culture conditions.

In the case of nodal explants as the source for our experiment, differences between particular averages were similar (Table 1). In summary, for two types of explant source-apical meristem and nodal segments-the number of new nodal segments depended on the presence of PGRs in the medium (Figure 1), with the best results observed for the MS-3 medium

In this case, similar results were observed for both types of explant. Higher values were obtained for lateral meristems, where the largest average minimum length of lateral shoots reached over 4.5 cm (MS-3) in the presence of auxin. Explants developed from apical meristems cultivated on the control medium were almost twice as low as those from lateral meristems. Excluding control media, the lowest values were observed for the MS-4 medium, which was supplemented only by cytokinin at 0.5 mg·dm^−3^ BA, regardless of the type of explant (Table 1).

The importance of choosing the right type of explant has been mentioned several times in research into mint. The experiment of Akter et al. (2016) [35] aimed to study the concentration of IBA for root regeneration, depending on the explant used. The nodal segment shoot tips and leaves were assessed at three IBA levels (1.0, 2.0, and 3.0 mg·dm^−3^) together with root regeneration control. The results revealed that the apical and lateral meristem performed better than the leaves in all combinations tested. It was shown that for in vitro regeneration of mint roots, the best results were obtained with 1.0 mg·dm^−3^ IBA with explants from apical or nodal segments. Akter and Hoque (2018) [29] conducted a similar experiment for the selected mint genotype (*Mentha* sp.), in which the same BAP concentrations and types of explants were used. The results showed much better shoot proliferation rates for apical and lateral meristems. Additionally, good results were obtained in our study for explants from an apical and lateral meristem. The same types of explant were also used in the experiment of Islam et al. (2017) [36], where ½ MS medium with the addition of different BAP and KIN concentrations was used. At a later stage of the experiment, the passage was changed to full MS medium with the addition of various concentrations of GA3 (gibberellic acid) to initiate elongation. The best results were obtained with a GA3 concentration of 1.0 mg·dm^−3^. In the next phase, rooting was carried out, which was the most effective on medium containing 1.5 mg·dm^−3^ IBA. Nodal explants were also used by Vasile et al. (2011) [37]. The highest rate of shoot regeneration was obtained on MS medium with the addition of 0.5 mg·dm^−3^ ZEA (zeatin) and 0.5 mg·dm^−3^ IAA, root regeneration on MS with 1.0 mg·dm^−3^ BAP and 1.0 mg·dm^−3^ IAA, while the elongation increase was most effective on MS with 0.5 mg·dm^−3^ ZEA and 0.5 mg·dm^−3^ IAA.

During our study of average root length, it was also noted that the lowest result for individual explants was obtained on MS-3 medium (0.5 mg·dm^−3^ IBA)-this result was less than half the length of the results for MS-1 and MS-4 when apical meristems were used, and also 3-5-fold less than the value for lateral meristems on MS-2 medium. Similar tendencies were also observed for explants from the apical meristem on MS-1 control medium comparing lateral meristems (Table 1).

In the case of apical meristems and nodal explants used as the initial material, the highest percentage of rooting explants was recorded on the control medium (100%), while only 37% and 83% of explants produced new lateral shoots on the control medium without PGRs, respectively (Figure 2). Rooting at the same or similar level compared to the control medium was observed in the explants with regenerated nodal segments on all tested media. The rooting of explants from the apical meristem after adding PGRs was the worst compared to rooted explants obtained from the nodal segment. These differences in rooting might have been due to the higher concentration of endogenous root-promoting substances in the nodal segments. In the case of our study, the addition of IBA did not result in the expected positive reaction in rooting, but we noticed a better elongation growth of shoots. The development of nodal segments resulted in the same level (100%) of lateral shoot development as the control (MS-K) on MS-3 medium (0.5 mg·dm^−3^ IBA) (Figure 3). The addition of PGRs to media caused a certain percentage of explants (%) to produce new shoots on MS-3 medium.

In our study, we did not identify callus formation, but the process of direct morphogenesis from lateral buds was noted. The average number of new lateral shoots was two. The length of the largest new lateral shoots was comparable to the plant height of explants derived from shoots with an apical meristem or single-nodal segments, but the smallest new lateral shoots were shorter than 1 cm. On average, we obtained five new segments per explant after four weeks of culture. The four variants of treatment with different combinations of PGRs enabled the development of an effective protocol for the rapid propagation of the tested peppermint genotype (Torseed Company, Toruń, Poland) from a shoot tip with an apical meristem and single-nodal segments. It was difficult to compare the impact of PGRs on the shooting and rooting process observed here and in other experiments, because the initial materials were different (other cultivars, clones from local origin, hybrids); however, some investigated traits, such as the length of shoots and roots and the number of new nodal segments, showed similar results to those reported elsewhere [1,37]. Optimal hormonal conditions for propagation differ from species to species, and also differ with the stage of development. The doses of growth regulators and combinations for our experiment were selected and adjusted based on the previously mentioned experiments; additionally, we introduced 0.5 mg·dm^−3^ 2iP (2-isopentinloadenine) + 0.1 mg·dm^−3^ IAA (MS-2), which turned out to be key to increasing the synthesis of interesting VOCs. This study was the first time that this cytokinin had been successfully used for these purposes in peppermint.

### 2.2. Peppermint’s Volatile Organic Compound (VOC) Composition

The chromatogram (Figure 3) shows the composition of the VOCs obtained from plants derived using various combinations of growth regulators. Overall, 53 VOCs were found, and 51 were identified (the mass spectra of unidentified VOCs are available in the Appendix A). There were no significant differences in the composition of any them; however, the type of plant hormones used affected the percentage of particular compounds in the VOC composition (Table 2). The VOCs of peppermint could represent up to 300 compounds, with major components such as menthol (30–55%) and menthone (14–32%). The Polish Pharmacopoeia VIII [39] determines the content of other ingredients as follows: cineole (3.5–14%), menthyl acetate (2.8–10%), isomenthone (1.5–10%), menthofuran (1.0–9.0%), limonene (1.0–5.0%), pulegone (<4.0%), and carvone (<1.0%). The quantitative composition of VOCs depends on many factors, such as in vivo and in vitro growing conditions, the date of harvest, post-harvest treatments, plant chemotypes, and others [40]. It should be emphasized that the chemotype of plants is extremely important, as Ludwiczuk et al. (2016) [41] showed in their studies focused on plants of *Mentha* species and their cultivars, where the range of possible differences in mint chemotypes was very wide. Some of them were characterized by higher amounts of limonene and 1,8-cineole (eucalyptol), which corresponded to the results of this study (limonene + eucalyptol 7.4–36.12%), but a difference was found in other major constituents which are expected to occur along with limonene and 1.8-cineole, such as carvone. This difference, as well as the lack of most characteristic peppermint VOCs—menthone, menthol, and their derivatives [42]—may have been caused by the very early stage of plant development in the in vitro culture. Figure 4 presents a chromatogram illustrating a comparison of peppermint VOC compositions depending on the source of the plant. Blue represents VOCs obtained from peppermint derived directly from in vitro culture from a medium without regulators. The sample marked in pink was made using plants subjected to acclimatization (in vivo). The green color shows the result obtained for mint plants after acclimatization, when earlier plants were cultivated on 0.5 IBA. The third option was tested because IBA had a greater effect on individual chemical compounds produced in vitro (Table 2). There were no differences in the qualitative compositions. The Santoro team (2013) [1] conducted similar research to maximize the growth and production of Eos in micropropagated peppermint seedlings. The experiment showed that supplementation with BAP alone resulted in the highest values of root length, root dry matter, shoot length, and number of nodes, leaves, and branches. The addition of IBA or IBA and BAP caused an approximately 50% increase in fresh shoot mass. The production of secondary metabolites such as menthone, menthol, menthofuran, and pulegone was influenced only by the addition of BAP, which resulted in a 40% increase in the total yield of EO. It has been established that the use of growth regulators increases the production of oils and biomass in herbaceous species rich in commercially valuable terpenes.

In Figure 5, the overall influence of growth regulators on the quantities of particular groups of peppermint VOCs is illustrated. The strongest influence was observed for VOCs occurring in small and moderate quantities (Figure 5A,B). For VOCs occurring in high amounts (Figure 5C), the variability was lower and was related mainly to menthofuranolactone, which, apart from limonene and 1.8-cineole, was the most abundant VOC in in-vitro-cultivated peppermints (24.69–67.27%). The literature review showed that this result was unique. Nevertheless, Frérot et al. (2002) [45] pointed out that *p*-menthane lactones such as menthofurolactone may have a significant impact on peppermint quality. Menthofurolactone in particular was characterized as having an interesting odor description: nicely coumarinic, phenolic, minty, and very powerful. The characteristic influence of menthafurolacton on aroma quality also was shown by Picard et al. (2017) [46] for red wines, where menthofurolactone, along with other *p*-menthane lactones, introduces freshness and minty nuances into the wine’s aroma profile. Regarding VOCs with a significant influence on peppermint’s odor (highlighted in Table 2), changes forced by applying growth regulators were observed for almost all OACs, except α-terpinene, *cis*-dihydro carvone, and carvone. The most noticeable changes were observed for the limonene and eucalyptol mixture and menthofurolactone.

Regarding the subject matter of this study, the most important aspect was the influence of applying growth regulators on the peppermint VOCs responsible for the plant’s OACs. According to Shigeto et al. (2019), Díaz-Maroto et al. (2008), and Frérot et al. (2002) [43,44,45], among VOCs identified in peppermint in this study, ethyl 2-methylbutyrate, α-pinene, 1-octen-3-ol, α-terpinene, eucalyptol, β-*trans*-ocimene, linalool, *cis*-dihydro carvone, carvone, piperitenone, and menthofurolactone may be considered OACs. As shown in Table 2, applying various growth regulators may result in different changes to peppermint’s VOC quantitative composition. This result suggests that there is a possibility of applying relevant growth conditions to adjust the aroma quality of peppermint for desired purposes. In particular, the MS-2 sample showed an interesting pattern in which the menthofuranolactone share was significantly increased, while the limonene and eucalyptol mixture share was decreased. In the face of other moderate changes in this sample, it would be possible to obtain products which are less herbaceous and balsamic, with a stronger minty aroma. Such an outcome might be a key point for peppermint cultivation strategies, and could be applied in cosmetics and pharmaceutical industries. In light of this study, it might be possible to use in vitro cultures supplemented with 0.5 mg·dm^−3^ 2iP + 0.1 mg·dm^−3^ IAA (MS-2) at a larger scale to obtain plants with an intense, clear, and fresh minty aroma; however, more tests, especially olfactometric and sensory ones, should be performed to confirm this. Additionally, further approaches to obtain other peppermint chemotypes with increased concentrations of compounds such as carvone, menthone, or menthol for pharmaceutical use should be considered.

The production of aromatic plant OACs may depend not only on genetic factors and plant development, but also on environmental factors. All of these can affect biochemical and physiological processes, which could modify the quantity and quality of VOCs. Currently, PGRs or plant hormones are used to affect EO production and chemical composition. Prins et al. (2010) [7] have studied the effect of PGRs and VOC production on many plants of aromatic use-herbaceous (peppermint, spearmint, sage, thyme etc.), shrubby (lavender, rose), and arborescent (spruce). Our experiment showed that the use of cytokines has an impact on the increase of peppermint EO production and the factor related to it, namely increased enzyme activity.

Considering that the content of peppermint VOCs depends on the regulator, the potential for modifying the volume of secondary metabolites produced is obvious. Plants treated with appropriate hormones can produce several times more volatile compounds. The experience of Tisserat and Vaughn (2008) [47] drew attention to the possibility of using growth regulators for commercial purposes. It is possible to modify not only the amount of chemical compounds contained, but also to eliminate undesirable substances, which was proven by the experiment of Bertoli et al. (2012) [4].

Tisserat and Vaughn (2000) [3] also conducted an experiment to test the influence of CO_2_ on VOC composition in mint (*Menth**a* L. spp.) and thyme (*Thymus vulgaris* L.). Although several terpenes occurred in the mint cultures in vitro, only two monoterpenes were conspicuous: limonene and piperitenone oxide. An increase in the CO_2_ levels did not appreciably increase limonene levels when shoots were grown on MS without sucrose. In contrast, limonene levels increased dramatically when shoots were cultured under 10,000 and 30,000 μmol·mol^−1^ CO_2_ levels on MS containing 3% sucrose. Piperitenone oxide levels increased to their highest levels at 3000 μmol·mol^−1^ CO_2_ for mint cultures grown on MS with and without sucrose. However, piperitenone oxide levels were markedly higher in cultures grown on MS with sucrose under ultra-high CO_2_ levels (i.e., ≥3000 μmol·mol^−1^ CO_2_) than in cultures grown on MS without sucrose. As much as 0.85 mg piperitenone oxide/g fresh weight of plant occurred in mint cultures grown on MS with sucrose under the 3000 μmol·mol^−1^ CO_2_ level. This was a 14-fold increase over the level found in cultures on MS with sucrose under ambient air. It should be noted that while the 3000 μmol·mol^−1^ CO_2_ level resulted in the highest growth and morphogenesis responses from mint plantlets in soil, it did not appreciably improve the levels of secondary metabolites. These results indicate that ultra-high CO_2_ levels may enhance the production of EOs and, consequently, VOC production in plants grown in vitro, meaning that they would be present at higher levels than in soil grown crops.

Tavares et al. (2004) [48] created an experiment to test the major VOCs of the linalool-producing *Lippia alba* Mill. N. E. Br. (Verbenaceae family) chemotype. Plantlets were grown in MS medium with or without PGR. The study showed that *L. alba* contains sabinene, myrcene, and linalool, which can also be found in peppermint. In this case, the addition of 0.23 µM of IAA to the medium significantly enhanced sabinene and myrcene contents, and the addition of 0.92 µM of KIN significantly increased the linalool level. These results may be useful to increase the levels of chemical compounds in peppermint.

Khanam and Mohammads’ experiment (2016) [46,49] was designed to study the effect of PGRs and to find the best ones for maximizing the efficiency and quality attributes of peppermint. The following regulators were used: BAP, CCC (chloromequat), GA_3_, IAA, IBA, KIN, MJ (methyl jasmonate), SA (salicylic acid), and Tria (triacontanol). Regulators were used at 60 days after transplanting (DAT) at concentrations of 5 × 10^−6^ M. The concentration, yield, and active constituents of peppermint were studied at 90 DAT. The data showed that SA was the best for increasing oil content and yield as well as menthol yield and concentration. The study identified three regulators with the greatest impact on menthol content-SA, GA_3_, and IBA. This experiment also proved that BAP was the best regulator for menthone content.

Stoeva and Iliev (1997) [50] studied the influence of phenylurea cytokinins (DROPP: 50% TDZ and 4PU-30–*N*-(2-chlor-4-pyridyl)-*N*‘-phenylurea) on the EO composition of a chosen cineole-type spearmint cultivar. The experiment showed that the use of cytokinins increased the major components (1,8-cineole and *p*-cymene) and reduced the amount of carvacrol and thymol nearly two-fold. It was concluded that the use of phenylurea cytokinins for this type of spearmint has a beneficial effect on EO productivity. For a well-balanced composition of EO, it is recommended to use 4PU-30 at 25 mg·dm^−3^ and DROPP at 100 mg·dm^−3^. At these concentrations of PGRs, the specific pharmacological activity should be guaranteed.

## 3. Materials and Methods

### 3.1. Micropropagation

Seeds of peppermint were used to start the culture. The seeds were purchased from the commercial company Torseed S.A. (https://torseed.pl/oferta/nasiona-ziol/). According to the manufacturer’s instructions, seeds were subjected to a process of stratification by placing them in the refrigerator for 7 days before the establishment of culture. Before proper disinfection, the seeds were skimmed and bled (treatment with a detergent solution followed by 70% alcohol). Proper disinfection was carried out in solutions of commercial Javel (1 volume of Javel: 3 volumes of sterile water) for 5, 8, or 12 min, and then they were washed three times in sterile water.

Prepared seeds were laid on the previously prepared MS medium (Murashige and Skoog, 1962) [51] and left to stand in the culture room (21 °C, photoperiod 16/8). After one month, 100% of the seeds had germinated and there were no signs of culture infection regardless of the disinfection time used. The experimental material consisted of peppermint microcuttings derived from shoot fragments which were cultivated in three passages on MS medium [48] without the addition of growth regulators. Plants were transferred to fresh medium every four weeks.

After the last passage, apical meristems with two leaves and nodal explants with two leaves were cultivated on fresh MS medium with four treatments of growth regulators: 0.5 mg·dm^−3^ BA (6-benzyloadeninopurine) + 0.1 mg·dm^−3^ IAA (indolyl-3-acetic acid) (MS-1), 0.5 mg·dm^−3^ 2iP (2-isopentinloadenine) + 0.1 mg·dm^−3^ IAA () (MS-2), 0.5 mg·dm^−3^ IBA (indolyl-3-butyric acid) (MS-3), 0.5 mg·dm^−3^ BA (MS-4), or in MS control medium without the addition of plant hormones (MS-K). Next, 30 g·dm^−3^ sucrose and 8 g·dm^−3^ agar were added to each medium and the pH was adjusted to 6.0 before autoclaving. The experiment was performed in triplicate. Ten explants for each replications of the experiment-apical meristem (Series I) or single-nodal segments (Series II)—were placed in a dedicated vessel containing media with the specified treatment. After four weeks, the characteristics were assessed: height of plant explants (cm), number of nodal segments, length of lateral shoots (cm; minimum and maximum), and length of roots (cm). In total, 30 plants in each medium variant were taken into account. Additionally, the percentage of explants with new shoots and roots was also calculated.

After the end of experiment, the plants were transferred to MS medium without the addition of growth regulators. After four weeks, all plants were acclimated.

### 3.2. Aroma Profiling

Plants cultivated on different media were subjected to headspace solid-phase microextraction (HS-SPME) with 2 cm DVB/CAR/PDMS fiber (Supeclo, Bellefonte, PA, USA). Briefly, 1 ± 0.05 g of leaves was placed in a 10 mL headspace vial with 10 μL of 2-undecanone (10,000 ppm) as an internal standard (IS). The vial was then placed on a heating plate at 70 °C for 5 min of conditioning, and the fiber was then exposed above the sample for 25 min. Analyte desorption was carried out for 2 min at 220 °C, split 1:100. The GC-MS analysis was carried out on a Varian CP-3800/Saturn 2000 (Varian, Walnut Creek, CA, USA) with a Zebron ZB-5 MSI column (30 m × 0.25 mm × 0.25 µm) (Phenomenex, Torrance, CA, USA). The temperature program started at 40 °C for 3 min, which increased to 110 °C at a rate of 5 °C/min and then to 270 °C at a rate of 20 °C/min. Scanning was carried out from 38 to 310 *m*/*z* at 70 eV. Helium was used as carrier gas at a flow rate of 1.0 mL/min. The procedure was repeated for plant material without the addition of growth regulators derived directly from the in vitro culture, without acclimatization and after acclimatization (two plant variants from the control medium). The analyses were carried out in triplicate.

Compounds were identified by comparing experimentally obtained linear retention indices (LRIs) calculated against a C6–C30 *n*-alkane mix (Sigma-Aldrich, Saint Louis, MO, USA), and mass spectra with those accessed in libraries (NIST 17 Mass Spectral and Retention Index Libraries (NIST17) and NIST WebBook) and the literature [52]. Compound semi-quantification was performed by normalizing the peak area and calculations based on the amount of added IS. For semi-quantification, no standard curves were used for any of the compounds found in the samples. IS concentrations were compared with those of the other compounds, by supposing a general equipment response factor of 1 for all volatile compounds.

### 3.3. Statistical Analysis

The results were expressed as the mean of the measurements and reported as mean ± SD (standard deviation). A two-way analysis of variance (ANOVA) was conducted to verify the lack of significance of PGRs (plant growth regulators), type of explant, and the interaction of explant type and the influence of PGR on peppermint micropropagation. The model of the two-way ANOVA is presented below:(1)Yij=μ+τi+βj+γij+εijkfor i=1, 2, …, a; j=1, 2, …, b; k=1, 2, …, r
where *Yil* is the mean value of the analyzed character, *μ* is the overall mean response, *τi* is the effect due to the *i*th level of factor A (medium); *βj* is the effect due to the jth level of factor B (type of explant), *γij* is the effect due to any interaction between the ith level of A and the *j*th level of B, and ε*ijk* is the random error component. The hypothesis assumed no impact of medium (supplemented in various PGRs), type of explant, or interactions on peppermint micropropagation. The significant differences were assessed at levels of 0.05 and 0.01. When an analysis of variance gave a significant result, Tukey’s HSD test was performed to compare mean values [53]. Data obtained during aroma profiling were subjected to a one-way analysis of variance using Tukey’s test (*p* < 0.05).

The obtained results were subject to statistical analysis with the use of the Statistica program, version 13.3. Tukey’s HSD test was performed in cases where the hypothesis was rejected.

## 4. Conclusions

This study on the impact of micropropagation conditions and cultivation process on peppermint volatile organic compound compositions revealed that the combination of growth regulators may have a noticeable influence on the plant’s aroma profile and growing efficiency. The applied combinations of growth regulators—0.5 mg·dm^−3^ BA + 0.1 mg·dm^−3^ IAA (MS-1), 0.5 mg·dm^−3^ 2iP + 0.1 mg·dm^−3^ IAA (MS-2), 0.5 mg·dm^−3^ IBA (MS-3), 0.5 mg·dm^−3^ BA (MS-4)—showed a capability to decrease the time required for plant multiplication.

Peppermint chemotype, used as an object of this study, was characterized by high amounts of limonene and eucalyptol (1,8-cineole) mixture and menthofurolactone, which was recently described as new *p*-menthane lactone that makes a significant contribution to the typical peppermint aroma, and a lack of menthone and menthol. One of the media used, containing 0.5 mg·dm^−3^ 2iP and 0.1 mg mg·dm^−3^ IAA, had a significant impact on the menthofurolactone share in the in-vitro-cultivated plants’ aroma profiles, evaluated using the HS-SPME technique. This result may suggest a promising approach for the usage of peppermint micropropagation with growth regulators to provide products with desired aroma profiles.

## Figures and Tables

**Figure 1 molecules-25-02652-f001:**
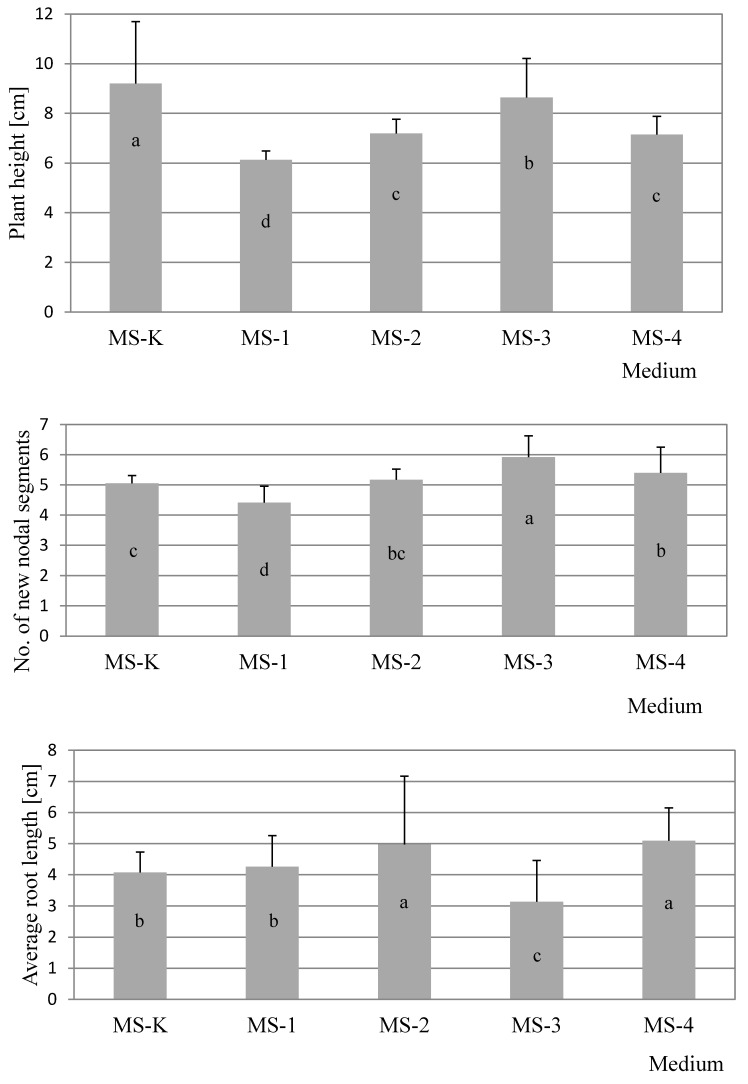
Significant effects of media on the three evaluated traits in developing mint explants. Columns followed by the same letter are not significantly different (*p* > 0.05, Tukey’s test) for a significant effect of the medium; bars represent standard deviations.

**Figure 2 molecules-25-02652-f002:**
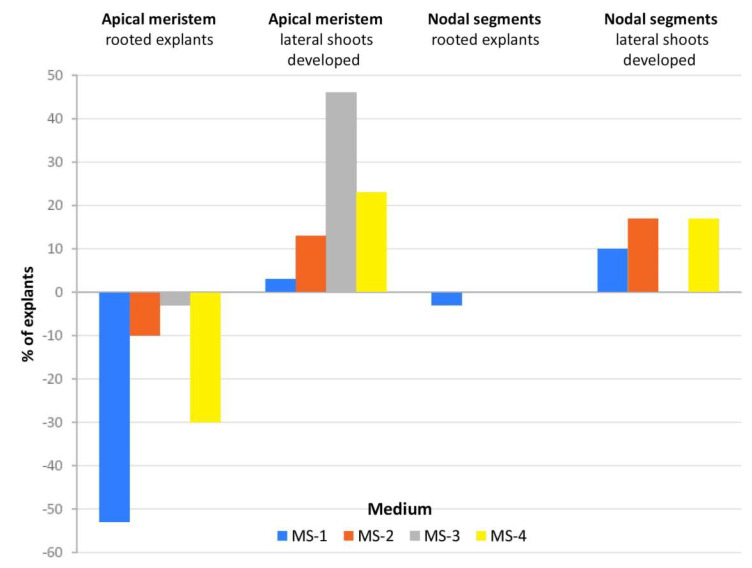
Developed roots and lateral shoots (%) of particular types of explant cultivated on four kinds of medium-MS-4, MS-3, MS-2, and MS-1-compared to control medium (without hormones).

**Figure 3 molecules-25-02652-f003:**
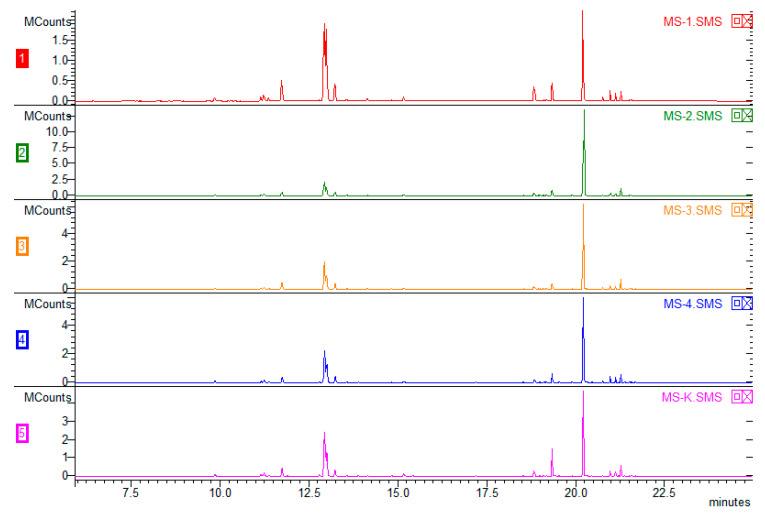
Comparison of the volatile organic compound chemical compositions of plants cultivated with various combinations of tested plant growth regulators: MS-1 (1), MS-2 (2), MS-3 (3), MS-4 (4), and without regulators (5) (MS-K).

**Figure 4 molecules-25-02652-f004:**
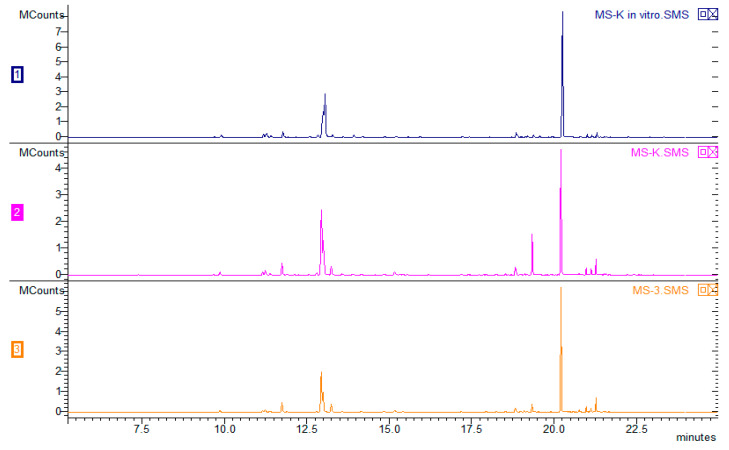
Chromatogram composition of peppermint VOCs directly from plants cultivated in an in vitro culture control (chromatogram 1–MS-K in vitro), with acclimatization control, after earlier cultivation on control medium (chromatogram 2–MS-K) and with acclimated cultivation earlier in tissue culture (chromatogram 3–MS-3).

**Figure 5 molecules-25-02652-f005:**
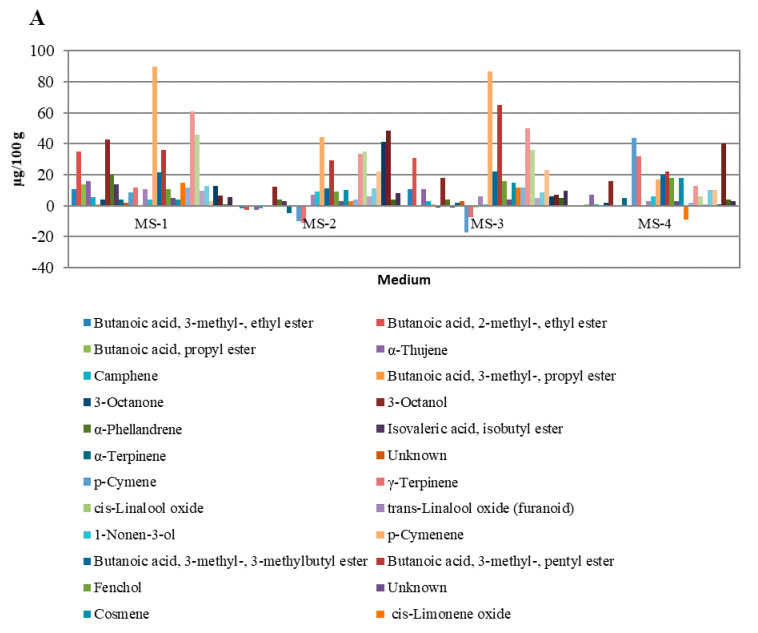
Differences in the content of individual VOC compounds extracted from plants cultivated on experimental media compared to MS-K medium without plant growth regulators. (Chart **A**): VOCs with concentrations of 0.2 to 100 µg/100 g of fresh leaves; (Chart **B**): VOCs with concentrations of 2.0 to 400 µg/100 g of fresh leaves; (Chart **C**): VOCs with concentrations greater than 0.4 mg/100 g of fresh leaves.

**Table 1 molecules-25-02652-t001:** Analyzed characteristics’ average values after four weeks of in vitro culture of initial explants of peppermint.

Medium	Explant Type	Plant Height (cm)	No. of New Nodal Segments	No. of New Lateral Shoots	Length of the Smallest Lateral New Shoots (cm)	Length of the Largest Lateral New Shoots	Average Root Length (cm)
MS-K	Apical meristem	11.5 ± 0.16 ^a^	5.1 ± 0.21 ^ns^	1.9 ± 0.12 ^ns^	2.2 ± 0.20 ^f^	5.0 ± 0.60 ^ns^	3.5 ± 0.15 ^e^
MS-1		5.8 ± 0.20 ^h^	4.1 ± 0.36 ^ns^	1.8 ± 0.54 ^ns^	2.3 ± 0.15 ^f^	2.8 ± 0.97 ^ns^	3.4 ± 0.24 ^e^
MS-2		6.7 ± 0.13 ^fg^	4.9 ± 0.06 ^ns^	1.8 ± 0.54 ^ns^	3.1 ± 0.24 ^bc^	4.1 ± 1.41 ^ns^	3.0 ± 0.29 ^e^
MS-3		10.1 ± 0.12 ^b^	5.5 ± 0.67 ^ns^	2.6 ± 0.57 ^ns^	2.6 ± 0.14 ^ef^	4.1 ± 0.29 ^ns^	1.9 ± 0.07 ^f^
MS-4		6. 5 ± 0.15 ^fg 1^	4.7 ± 0.57 ^ns 2^	2.2 ± 0.73 ^ns^	2.2 ± 0.19 ^f^	3.3 ± 1.26 ^ns^	4.2 ± 0.17 ^d^
Average		8.1 ± 2.31 ^a^	4.9 ± 0.61 ^b^	2.1 ± 0.54 ^ns^	2.5 ± 0.40 ^b^	3.9 ± 1.13 ^ns^	3.4 ± 0.77 ^b^
MS-K	Nodal segment	6.9 ± 0.23 ^ef^	5.0 ± 0.32 ^ns^	1.4 ± 0.29 ^ns^	4.4 ± 0.13 ^a^	4.6 ± 0.23 ^ns^	4.7 ± 0.23 ^cd^
MS-1		6.4 ± 0.10 ^g^	4.7 ± 0.55 ^ns^	2.3 ± 1.01 ^ns^	3.3 ± 0.09 ^bc^	4.5 ± 0.38 ^ns^	5.2 ± 0.14 ^c^
MS-2		7.7 ± 0.25 ^cd^	5.5 ± 0.21 ^ns^	2.9 ± 0.81 ^ns^	3.6 ± 0.10 ^b^	5.2 ± 0.75 ^ns^	7.0 ± 0.23 ^a^
MS-3		7.2 ± 0.13 ^de^	6.3 ± 0.61 ^ns^	1.4 ± 0.30 ^ns^	4.5 ± 0.20 ^a^	4.9 ± 0.60 ^ns^	4.3 ± 0.28 ^d^
MS-4		7.8 ± 0.15 ^c^	6.1 ± 0.40 ^ns^	2.2 ± 0.78 ^ns^	2.8 ± 0.12 ^de^	4.6 ± 0.88 ^ns^	6.0 ± 0.31 ^b^
Average		7.2 ± 0.54 ^b^	5.5 ± 0.73 ^a^	2,0 ± 0.84 ^ns^	3.7 ± 0.68 ^a^	4.8 ± 0.58 ^ns^	5.4 ± 1.01 ^a^

^1^ Values followed by the same letter within a column are not significantly different (*p* > 0.05, Tukey’s test) for interaction, type of explant used, and medium, respectively; ^2^ ns: not significant;

**Table 2 molecules-25-02652-t002:** The composition of peppermint oil depending on the composition of the medium.

Compound	LRI_exp_ ^1^	LRI_lit_ ^2^	MS-K	MS-1	MS-2	MS-3	MS-4	Odor Description ^4^
			(mg/100 g ^3^)					
Butanoic acid, 3-methyl-, ethyl ester	857	849	tr ^5^	0.013	tr	0.014	tr	
Butanoic acid, 2-methyl-, ethyl ester	861	851	0.011 ^a 6^	0.045 ^b^	tr	0.041 ^b^	0.011 ^a^	fruity, strawberry-like
Butanoic acid, propyl ester	899	899	tr	0.013	tr	- ^7^	tr	
α-Thujene	930	929	0.011	0.026	tr	0.021	0.018	
α-Pinene	932	937	0.112 ^a^	0.282 ^c^	0.196 ^e^	0.331 ^b^	0.239 ^d^	terpene-like, pinene-like
Camphene	953	952	tr	tr	tr	tr	tr	
Sabinene	976	974	0.118	0.314	0.191	0.314	0.247	
β-Pinene	979	979	0.198	0.509	0.334	0.524	0.445	
1-Octen-3-ol	982	980	0.060 ^a^	0.276 ^b^	0.079 ^ad^	0.138 ^c^	0.095 ^d^	mushroom-like
3-Octanone	987	986	tr	tr	-	-	tr	
β-Myrcene	992	991	0.449	1.771	0.915	1.897	0.947	
3-Octanol	997	994	0.025	0.066	0.038	0.045	0.042	
α-Phellandrene	1004	1005	tr	0.028	0.015	0.014	tr	
Isovaleric acid, isobutyl ester	1006	1005	tr	0.019	tr	tr	tr	
α-Terpinene	1018	1017	0.010 ^a^	0.013 ^a^	tr	0.014 ^a^	0.016 ^a^	floral
Unknown	1021		tr	tr	-	tr	-	
*p*-Cymene	1025	1025	0.071	0.079	0.062	0.055	0.116	
Limonene+Eucalyptol	1031	1030	4.432 ^a^	6.030 ^b^	2.065 ^d^	7.703 ^c^	6.574 ^b^	herbal, camphor, minty, balsamic, eucalyptus
β-*cis*-Ocimene	1041	1038	0.357	1.385	0.783	1.531	1.018	
β-*trans*-Ocimene	1052	1049	0.043 ^a^	0.160 ^b^	0.088 ^d^	0.162 ^b^	0.118 ^c^	medicinal
γ-Terpinene	1062	1060	0.039	0.049	0.029	0.031	0.071	
*cis*-Sabinene hydrate	1070	1070	0.033	0.203	0.082	0.148	0.055	
*cis*-Linalool oxide	1075	1074	tr	-	tr	-	tr	
*trans*-Linalool oxide (furanoid)	1083	1085	tr	0.013	0.012	tr	tr	
1-Nonen-3-ol	1083	1080	tr	tr	0.012	tr	tr	
*p*-Cymenene	1090	1090	0.039	0.128	0.085	0.124	0.055	
Linalool	1099	1099	0.148 ^a^	0.397 ^b^	0.302 ^d^	0.348 ^c^	0.245 ^e^	floral, citrus-like
Butanoic acid, 3-methyl-, 3-methylbutyl ester	1099	1104	0.020	0.041	0.032	0.041	0.039	
Butanoic acid, 3-methyl-, pentyl ester	1099	1100	0.028	0.062	0.059	0.093	0.050	
Fenchol	1114	1116	0.013	0.024	0.023	0.028	0.032	
Unknown	1121		tr	tr	tr	tr	tr	
Cosmene	1127	1131	tr	0.011	0.021	0.024	0.026	
*cis*-Limonene oxide	1139	1134	0.017	0.032	0.021	0.031	tr	
*trans*-Verbenol	1152	1148	tr	0.015	tr	0.017	tr	
α-Terpineol	1177	1177	0.026	0.085	0.059	0.076	0.037	
Terpinen-4-ol	1183	1189	0.020	0.047	0.038	0.038	tr	
*cis*-Dihydro carvone	1194	1192	tr	0.015 ^a^	0.012 ^a^	0.010 ^a^	tr	sweet spices
Myrtenal	1196	1193	0.010	0.013	0.015	0.010	0.011	
*trans*-Isopiperitenol	1204	1210	0.015	0.017	0.038	0.038	0.024	
Cumic aldehyde	1229	1239	0.020	0.032	0.062	0.024	0.021	
Butanoic acid, 3-methyl-, 3-cis-hexenyl ester,	1240	1238	0.026	0.032	0.076	0.034	0.066	
Butanoic acid, 3-methyl-, hexyl ester	1245	1244	tr	tr	tr	tr	tr	
Carvone	1253	1247	tr	0.011 ^a^	0.015 ^a^	0.014 ^a^	tr	fresh, minty, herbal
*cis*-Carvone oxide	1261,	1263	0.348 ^a^	1.280 ^b^	0.713 ^d^	0.824 ^c^	0.600 ^e^	minty spearmint
*cis*-2-decen-1-ol	1271	1271	0.025	tr	0.053	0.145	0.026	
Perillal	1277	1272	0.027	0.058	0.144	0.076	0.068	
Pentanoic acid, 3-cis-hexenyl ester,	1282	1281	0.018	0.060	0.126	0.062	0.074	
Piperitenone	1339	1340	0.016 ^a^	0.030 ^a^	0.120 ^b^	0.055 ^c^	0.068 ^c^	cumin, anise-like
Menthofurolactone	1376	1367	3.845 ^a^	5.276 ^e^	19.727 ^b^	15.903 ^c^	11.753d	coumarin, phenolic, minty, very powerful
Caryophyllene	1415	1419	0.032	0.145	0.091	0.221	0.087	
*cis*-Muurola-4(15),5-diene	1464	1463	0.148	0.397	0.396	0.528	0.484	
Germacrene D	1481	1481	0.139	0.303	0.343	0.503	0.442	
Valencene	1499	1496	0.271	0.541	0.806	1.193	1.011	
			11.220 ± 0.428	20.344 ± 0.606	28.276 ± 1.038	33.445 ± 1.134	25.232 ± 0.989	
			0.303	0.452	0.707	0.778	0.47	

^1^ Experimentally obtained retention indices calculated against *n*-alkanes; ^2^ retention indices according to NIST17 database; ^3^ fresh leaf mass; ^4^ odor description of odor-active compounds (OACs) specified in the literature [43,44,45]; ^5^ trace < 0.010 mg/100 g; ^6^ values followed by the same letter within a row are not significantly different (*p* > 0.05, Tukey’s test); ^7^ “-“ compound not detected.

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
