# Peer review of "Mentha piperita L. Micropropagation and the Potential Influence of Plant Growth Regulators on Volatile Organic Compound Composition"

_molecules, 2020, doi:10.3390/molecules25112652_

Round 1

Reviewer 1 Report

In introduction, detailed description of micropropagation is lacking.  For this reason, it is hard to understand the experimental condition and result section.  Although authors examined the effect of supplements on the plant morphology and VOCs in leaf, I failed to understand why authors examined the effect of the IAA, 2iP, IBA etc. and their concentrations.  For this reason, I failed to understand the novelty and importance of this paper.  Indeed, authors state that many researchers report the effect of supplement including IAA, IBA on the protoplast culture.  The author should clearly and briefly state the background and issues of the research area, the purpose and the results.

In Fig. 5, I failed to understand that concentrations of some VOCs reduced to -mg/100g. What is -mg/100g?

Author Response

Dear Reviewers,

Thank you for your valuable comments regarding manuscript molecules-788520, entitled Mentha piperita L. micropropagation and potential influence of plant growth regulators on volatile organic compounds composition. We appreciate your detailed review and hope that our corrections will find your acceptance.

Reviewer I:

  1. In introduction, detailed description of micropropagation is lacking. For this reason, it is hard to understand the experimental condition and result section.  Although authors examined the effect of supplements on the plant morphology and VOCs in leaf, I failed to understand why authors examined the effect of the IAA, 2iP, IBA etc. and their concentrations.  For this reason, I failed to understand the novelty and importance of this paper.  Indeed, authors state that many researchers report the effect of supplement including IAA, IBA on the protoplast culture.  The author should clearly and briefly state the background and issues of the research area, the purpose and the results.

Thank you for this comment. More comprehensive background of this research was described with details in Introduction section (lines 88-128). Issues of growth regulators selection, concentration and usefulness regarding particular peppermint chemotype were addressed.

  1. In Fig. 5, I failed to understand that concentrations of some VOCs reduced to -mg/100g. What is -mg/100g?

Figure 5 do not present total amount of particular VOCs in the aroma profile of peppermint regarding specific treatments, but presents the shift in particular VOCs concentrations between samples cultivated with growth regulators and control medium. Our intention is to illustrate the change in VOCs concentrations forced by applying growth regulators. For instance, on Figure 5C, one may observe that the concentration of limonene+eucalyptol in sample MS-2 was lower approximately 2.5 mg per 100 g of fresh leaves than in sample MS-K (control medium)

Reviewer 2 Report

The authors presented a research article about Mentha piperita L. micropropagation and potential influence of plant growth regulators on volatile organic compounds composition.

The topic is interesting and well within the aims and scopes of the Journal Special Issue.

Yet, I deem the manuscript needs some changes and implementations, especially, in the discussion section.

For this reason, I recommend a Major Revision.

The things to do and check are listed below one by one:

ABSTRACT:

- Line 13: “Mentha piperita L. belongs to herbaceous plants…” What do you exactly mean with this? There are so many herbaceoous plant in the world. Please be more specific, for example, citing the family this family belongs to.

- Line 16: In vitro should be written in Italics.

- “…for obtaining more active substances contained in the plant than the traditional cultivation.” Active in what? Moreover, I remind he authors that this can also be achieved without in vitro culture, e.g. by playing with the environmental conditions and so on. Hence, this sentence is not so correct. Please modify it accordingly.

- “lateral shoots were induced in 83% in both variants…” Induced in 83% in both variants? What do you exactly mean?

- “…and the content of active substances in the plant increased significantly.” Content in what context? Please specify.

- Line 24: In vitro should be written in Italics.

KEYWORDS:

- The keyword in vitro is not fine as it is. Please modify or erase it.

INTRODUCTION:

- “Biotechnology tools such as in vitro cultures allow obtaining unified plant material with the best quality characteristics for the cultivation of herbaceous plants…” Best quality characteristics for the cultivation of herbaceous plants? Why?

- “… to obtain seedlings with the characteristics…” What kind of characteristics? This is too generic now and it seems it is for everything whereas it is not. Please specify.

- “Furthermore, as characteristic for peppermint 1,8-cineole, limonene, trans-menthyl acetate, trans-sabinene hydrate, peppermint EO…” What do you mean with the first part of this sentence? I think I can understand the meaning but please rephrase it anyway for a better understanding.

- “… to have health-promoting properties…” I would use the term beneficial pharmacological rather than health-promoting .

-  Line 64: In vitro should be written in Italics.

- I have a question. Has this kind of study or similar study ever been done before? If so, for what purpose, in what context? Please cite these studies. If not, please write something about the importance of this study in the field specifying why you used peppermint. This part is useful to draw more readers.

RESULTS:

- “Type of starting material to determine the experiments was significant for most of the analyzed characters, without length of the largest lateral new shoots and number of new lateral shoots”. What does this sentence exactly mean? Please rephrase it better. Do you mean without considering the length of the largest lateral new shoots and the number of new lateral shoots?

- Caption of Table 1: In vitro should be written in Italics. The Table format and style should be the same in every part. Please level them.

- Lines 98-105: How do you explain these results? What did you do in order to obtain them? And, in comparison with the other results, what are the differences? You must discuss about all these points.

- Lines 154-156: How do you explain these results?

- Line 163: Please modify the relative part as follows: “…mass spectra of unidentified…”.

- Line 172: Please modify the relative part as follows: “There were no differences in the qualitative composition” Why this, in your opinion? Please discuss a little about this.

- Table 2: Please erase the column with the CAS numbers which is useless as well as the color of the odor description. Indeed, present a table with the Mass data. Why the line limonene + eucalyptol together? In the line for cis-limonene oxide, eliminate the comma. I did not notice menthol among the compounds? Is it right? Mg/100 g of what? Extract? Please specify.

- Line 193: The right term is illustrated.

- Lines 195-199: Why these results? How do you explain them?

DISCUSSION:

- “To optimize the regeneration process from different initial peppermint explants, mainly PGRs such as BA, BAP, TDZ, KIN and 2.4D, NAA, IAA or IBA” Please specify these symbols for all the readers.

- Line 220: In vitro should be written in Italics.

- Line 296: In vitro should be written in Italics.

- “The quantitative composition of VOCs depends on many factors, such as in vivo and in vitro growing conditions, and the date of the harvest.” Not only. What about the others?

- Line 318: “As may be observed in Table..” Table?

- “…may force different changes in peppermint’s VOCs composition.” Composition is too generic? You must write quantitative composition since, as you wrote, the qualitative one is the same.

- Line 322: The right term is eucalyptol.

- Line 342: The right terminology is Mentha L. spp.

- Line 359: Lippia alba. Please write the complete botanical name of the species.  

- Please merge this part with the RESULTS section placing the relative part of the discussion near the relative part of the results. This is better.    

- I think you should better develop the point about how useful a major amount of VOCs can be in practice in your context. I mean, why do this for peppermint? What would you do with these new things? A small discussion about the link between VOCs and the pharmacological and aromatic properties would be appreciated and would make this paper more interesting.

- I might be fine with all the examples you wrote about in this section for showing the importance of choosing the best growth regulators and media. Yet, I deem a discussion about the comparison of your results with others as reported in the literature concerning your same plant and growth regulators is necessary, if there are any more. This is not clear. If not, you must specify this.

MATERIALS AND METHODS:

- What is Torseed Company? Why is it at that point?

- Please specify where the peppermint seeds derive from.

- Line 422: In vitro should be written in Italics.

CONCLUSIONS:

- “Peppermint, used as an object of this study, represented a chemotype characterized by…”  What? What’s this?

Author Response

Dear Reviewers,

Thank you for your valuable comments regarding manuscript molecules-788520, entitled Mentha piperita L. micropropagation and potential influence of plant growth regulators on volatile organic compounds composition. We appreciate your detailed review and hope that our corrections will find your acceptance.

Reviewer II:

  1. Line 13: “Mentha piperita L. belongs to herbaceous plants…” What do you exactly mean with this? There are so many herbaceoous plant in the world. Please be more specific, for example, citing the family this family belongs to.

Thank you for this point. The sentence was rephrased for: " Mentha piperita L. belongs to Lamiaceae family and is […]”

  1. Line 16: In vitro should be written in Italics.

This has been corrected.

  1. “…for obtaining more active substances contained in the plant than the traditional cultivation.” Active in what? Moreover, I remind he authors that this can also be achieved without in vitro culture, e.g. by playing with the environmental conditions and so on. Hence, this sentence is not so correct. Please modify it accordingly.

Thank you for this comment. The sentence was rephrased for:Moreover, in vitro cultures may allow to affect the production of desirable odour-active compounds (OACs) and improve their share in plant aroma profile.”

  1. “lateral shoots were induced in 83% in both variants…” Induced in 83% in both variants? What do you exactly mean?

In this case, a very high degree of rooting was found (97% for apical meristems, 100% for nodal meristems), lateral shoots were induced in 83% in both type of explants.

  1. “…and the content of active substances in the plant increased significantly.” Content in what context? Please specify.

Thank you for this point. We have provided more specific information: “…and the content of OACs in the plant aroma profile increased significantly, especially menthofurolactone and cis-Carvone oxide, responsible in this case for characteristic mint-like aroma.”

  1. Line 24: In vitro should be written in Italics.

This has been corrected.

  1. The keyword in vitro is not fine as it is. Please modify or erase it.

This has been corrected.

  1. “Biotechnology tools such as in vitro cultures allow obtaining unified plant material with the best quality characteristics for the cultivation of herbaceous plants…” Best quality characteristics for the cultivation of herbaceous plants? Why?

We have rephrased this sentence and provided more specific characteristic of “best quality” meaning: “Biotechnology tools such as in vitro cultures, are useful for obtaining unified plant material with the guaranteed aroma quality since the cultivation physical and chemical conditions may be strictly controlled, including temperature, photoperiod  basal medium, addition of plant growth regulators and other factors as CO2.”

  1. “… to obtain seedlings with the characteristics…” What kind of characteristics? This is too generic now and it seems it is for everything whereas it is not. Please specify.

We have erased this statement.

  1. “Furthermore, as characteristic for peppermint 1,8-cineole, limonene, trans-menthyl acetate, trans-sabinene hydrate, peppermint EO…” What do you mean with the first part of this sentence? I think I can understand the meaning but please rephrase it anyway for a better understanding.

Thank you for this comment. The sentence was rephrased for: “Furthermore, characteristic for peppermint EO VOCs, like 1,8-cineole, limonene, trans-menthyl acetate, trans-sabinene hydrate, is appreciated  in the cosmetics and pharmaceutical industries for its refreshing aroma and bioactive features.”

  1. “… to have health-promoting properties…” I would use the term beneficial pharmacological rather than health-promoting .

This has been corrected.

  1. Line 64: In vitro should be written in Italics.

This has been corrected.

  1. I have a question. Has this kind of study or similar study ever been done before? If so, for what purpose, in what context? Please cite these studies. If not, please write something about the importance of this study in the field specifying why you used peppermint. This part is useful to draw more readers.

Thank you for this suggestion. We have introduced detailed background for this research. Please see lines 88-121.

  1. “Type of starting material to determine the experiments was significant for most of the analyzed characters, without length of the largest lateral new shoots and number of new lateral shoots”. What does this sentence exactly mean? Please rephrase it better. Do you mean without considering the length of the largest lateral new shoots and the number of new lateral shoots?

This sentence was rephrased on: “Conducted analysis of variance showed, that type of starting material was significant for most of the analyzed characters, without length of the largest lateral new shoots and number of new lateral shoots and mean values were grouped using Turkey’s test…”

  1. Caption of Table 1: In vitro should be written in Italics. The Table format and style should be the same in every part. Please level them.

This has been corrected.

  1. Lines 98-105: How do you explain these results? What did you do in order to obtain them? And, in comparison with the other results, what are the differences? You must discuss about all these points.

This has been corrected and explanations for these results have been added in lines 150-155. Moreover the discussion was updated in lines 166-185

  1. Lines 154-156: How do you explain these results?

Explanations for these results have been added in lines 252-256.

  1. Line 163: Please modify the relative part as follows: “…mass spectra of unidentified…”.

This has been corrected.

  1. Line 172: Please modify the relative part as follows: “There were no differences in the qualitative composition” Why this, in your opinion? Please discuss a little about this.

This has been corrected. Moreover the discussion was updated in lines 292-300.

  1. Table 2: Please erase the column with the CAS numbers which is useless as well as the color of the odor description. Indeed, present a table with the Mass data. Why the line limonene + eucalyptol together? In the line for cis-limonene oxide, eliminate the comma. I did not notice menthol among the compounds? Is it right? Mg/100 g of what? Extract? please specify.

Thank you for this comment. The changes in the table have been made and the details regarding concentration units have been given.

The limonene and eucalyptol are given together, since the compounds were not completely separated during GC-MS analysis. In this case, we decided to give the amount of this two VOCs as sum, than risk improper, particular amounts estimation.

That is true that menthol was not detected. We justify this result by occurrence of various mint chemotypes, which may differ in VOCs. In this case we think that used in the study peppermint was characterized by high amounts of menthofurolactone, limonene and eucalyptol and by lack of menthol. We discuss this result more in lines 278-285.

  1. Line 193: The right term is illustrated.

This has been corrected.

  1. Lines 195-199: Why these results? How do you explain them?

Please find the discussion regarding this results in lines 337-354.

  1. “To optimize the regeneration process from different initial peppermint explants, mainly PGRs such as BA, BAP, TDZ, KIN and 2.4D, NAA, IAA or IBA” Please specify these symbols for all the readers.

This has been corrected. Particular abbreviations have been explained.

  1. Line 220: In vitro should be written in Italics.

This has been corrected.

  1. Line 296: In vitro should be written in Italics.

This has been corrected.

  1. “The quantitative composition of VOCs depends on many factors, such as in vivo and in vitro growing conditions, and the date of the harvest.” Not only. What about the others?

Thank you for this point. The list of factors was expanded witch such examples as post-harvest treatments and chemotypes. Of course those are just an examples of factors, however we find them as the most important for VOCs characteristics.

  1. Line 318: “As may be observed in Table..” Table?

This has been corrected for Table 2.

  1. “…may force different changes in peppermint’s VOCs composition.” Composition is too generic? You must write quantitative composition since, as you wrote, the qualitative one is the same.

This has been corrected.

  1. Line 322: The right term is eucalyptol.

This has been corrected.

  1. Line 342: The right terminology is Mentha L. spp.

This has been corrected.

  1. Line 359: Lippia alba. Please write the complete botanical name of the species.

This has been corrected.

  1. Please merge this part with the RESULTS section placing the relative part of the discussion near the relative part of the results. This is better.

Thank you for this suggestion. The Results and Discussion sections have been combined.

  1. I think you should better develop the point about how useful a major amount of VOCs can be in practice in your context. I mean, why do this for peppermint? What would you do with these new things? A small discussion about the link between VOCs and the pharmacological and aromatic properties would be appreciated and would make this paper more interesting.

Thank you for this suggestion. The appropriate discussion part was added. Please see lines 348-355.

  1. I might be fine with all the examples you wrote about in this section for showing the importance of choosing the best growth regulators and media. Yet, I deem a discussion about the comparison of your results with others as reported in the literature concerning your same plant and growth regulators is necessary, if there are any more. This is not clear. If not, you must specify this.

Thank you for this suggestion. Please see lines 395-412.

  1. What is Torseed Company? Why is it at that point?

Seeds of peppermint were used to start the culture. The seeds were purchased from a commercial company called Torseed.

  1. Please specify where the peppermint seeds derive from.

The seeds were purchased from a commercial company Torseed S.A. (https://torseed.pl/oferta/nasiona-ziol/).

  1. Line 422: In vitro should be written in Italics.

This has been corrected.

  1. “Peppermint, used as an object of this study, represented a chemotype characterized by…” What? What’s this?

The sentence was rephrase for: “Peppermint chemotype, used as an object of this study, was characterized by high amounts of limonene and eucalyptol (1,8-cineole) mixture and menthofurolactone, which has recently been described as new p-menthane lactone with significant contribution in typical peppermint aroma and lack of menthone or menthol.”

Our intention was to underline, that the VOCs composition of plant used in the study, is strictly bounded with the plant chemotype. Chemotype, which in this case is characterized by high amounts of limonene, eucalyptol and menthofurolactone and lack of menthol or menthone.

Round 2

Reviewer 1 Report

Lrystian et al. investigated the effect of plant growth regulators (PGRs) on the in vitro culture peppermint initial and it was found that supplementation of indolyl-3-butyric acid (IBA) is  most effective. However, the writing is awkward and lacks conciseness. Further, the writing often lacks clarity and sharpness, and result sections are poorly organized. Information of ref. in result section is too long. Authors should unify the unit to mg dm-3 or mg / L. In addition, it was failed to explain why authors used the 4 treatments of growth regulators including BA, IAA, IBA at the specified concentrations. Why authors did not use other regulators? Authors should unify the expression, especially PGRs, auxin,  cytokines and hormones.  In Fig. 4, chromatogram shown in blue is lacking. I cannot understand the ''- XX ug/100 g"" or ''-mg/100 g''.

Author Response

Dear Reviewer,

Thank you for your valuable comments regarding manuscript molecules-788520, entitled Mentha piperita L. micropropagation and potential influence of plant growth regulators on volatile organic compounds composition, in second round. We appreciate your detailed review and hope that our statements given in pdf file will find your acceptance.

Jacek Łyczko, corresponding author

Reviewer 2 Report

The authors presented a revised version of the manuscript I have previously reviewed.

The authors satisfactorily addressed all my most important queries.

Yet, there are still a couple of minor writing mistakes such as:

- Lines 151: The phrase should be “Cytokines generally inhibit elongation…”.

- Line 273: The right term is “determines”.

These things may be even corrected during the Proofreading step.

For this reason, I recommend its acceptance for publication in this Journal in its present form.

Author Response

Dear Reviewer,

thank you for your kind opinion and manuscript acceptance. We are glad, that our corrections have satisfy you.

Kind regards,

Jacek Łyczko, corresponding author